# Universal or Personalized Mesenchymal Stem Cell Therapies: Impact of Age, Sex, and Biological Source

**DOI:** 10.3390/cells11132077

**Published:** 2022-06-30

**Authors:** Diana M. Carp, Yun Liang

**Affiliations:** Department of Medical Microbiology and Immunology, University of Wisconsin-Madison, Madison, WI 53706, USA; dmcarp@wisc.edu

**Keywords:** autoimmune disease, stem cell therapy, mesenchymal stem cells, age, sex

## Abstract

Mesenchymal stem/stromal cells (MSCs) hold great promise for the treatment of autoimmune conditions given their immunomodulatory properties. Based on the low immunogenicity of MSCs, it is tempting to consider the expansion of MSCs from a “universal donor” in culture prior to their allogeneic applications for immediate care. This raises the critical question of the criteria we should use to select the best “universal donor”. It is also imperative we compare the “universal” approach with a “personalized” one for clinical value. In addition to the call for MHC-matching, recent studies suggest that factors including age, sex, and biological sources of MSCs can have significant impact on therapy outcome. Here, we will review findings from these studies, which shed light on the variables that can guide the important choice of “universal” or “personalized” MSC therapy for autoimmune diseases.

## 1. Introduction

Mesenchymal stem/stromal cells (MSCs), originally identified in mouse bone marrow, are currently defined as plastic-adherent, fibroblast-like cells that carry a defined set of markers and can differentiate into cells of multiple lineages, including osteoblasts, adipocytes and chondroblasts, in vitro [1,2,3]. In addition to their multipotent properties, MSCs produce paracrine factors in the form of cytokines and growth factors, as well as extracellular vesicles [4,5,6]. These paracrine factors have immunomodulatory functions, including the inhibition of B- and T- cell proliferation and monocyte maturation, and the promotion of regulatory T cell and M2 macrophage generation [4,5,7,8]. Because MSCs can be home to inflammatory sites where they exert their immunosuppressive function, they hold great therapeutic promise in the modulation of inflammatory and autoimmune diseases.

Autoimmune diseases occur when the body’s immune system attacks itself and healthy cells, tissues, and organs [9]. Around the world, there has been an increase in the prevalence of autoimmune diseases from the 4% estimated in the early 2000s to a disturbing 9%. In the United States alone, between 5 and 8% of the population suffer from autoimmune diseases, and it has become a growing concern due to its heritability. There are more than 80 autoimmune diseases that have been discovered and are recognized by the NIH and other health institutes around the world. Currently there is no cure for autoimmune diseases, and there has been a growing interest in the potential use of MSCs as disease therapy. It is estimated that by June 2020, there will have been 1138 clinical trials involving human MSCs, with a majority of them in their second phase [10].

In the therapeutic exploration of MSCs, one critical question is the choice of autologous versus allogeneic transfer. MSCs have long been considered hypoimmunogenic, which can potentially enable MSC transplantation across major histocompatibility barriers and cell expansion in culture prior to transfer [1,11]. Indeed, adult MSCs are being investigated to treat a wide range of diseases including lupus, arthritis, multiple sclerosis, Crohn’s disease, graft versus host disease, myocardial infarction, stroke, acute lung injury, chronic obstructive pulmonary disease, cirrhosis, amyotrophic lateral sclerosis and diabetes, with the majority of patients receiving allogeneic MSCs [12,13]. Allogeneic MSC therapy is generally considered safe, with the potential caveat that most studies do not characterize if donors and recipients are MHC-matched or mismatched [14].

One major benefit of allogeneic MSC therapy is that it allows for cell expansion in culture prior to clinical use. In other words, one can have enough cells, which can take weeks to obtain from culture, for immediate administration at the time of diagnosis. In addition, allogeneic transfer enables the use of MSCs from healthy subjects, which may have significant phenotypic differences from patient MSCs, a consideration especially important for autoimmune diseases. Indeed, many companies driving clinical trials exploring MSC therapies derive MSCs from a small number of donors and expand cells for the treatment of entire cohorts of patients [1].

The potential approach of “universal donor” MSCs raises the critical question of the criteria we should use to select for the best “universal donor”. The “universal donor” refers to the use of MSCs from one donor to transplant into multiple patients, as opposed to a complete “personalized” approach to derive a unique and specific MSC line for each patient. It is also important that we evaluate its clinical value compared to a personalized approach. In addition to the call for MHC-matching, recent studies suggest that factors including age, sex, and biological sources of MSCs can have a significant impact on therapy outcome. Here, we will review findings from these studies with a focus on autoimmune diseases, which shed light on the variables that can guide the important choice of “universal” or “personalized” MSC therapy for autoimmune disease patients.

## 2. Impact of Age on the Immune System and MSCs

In MSC transplantation, both the immune system of the recipient and the donor MSCs, are impacted by age. Both aspects should be considered when developing MSC therapies for autoimmune disease patients, which will be discussed in this section.

### 2.1. Aging of the Immune System

With increased aging, there are modifications to both innate and adaptive immunity. The notions of “immunosenescence” or “inflammaging” are relatively new concepts that have been adopted in the immunology and aging communities to refer to the aging immune system and properties associated with inflammation. Coined in 1969 by Roy L. Walford, the autoimmune theory of aging refers to immune ineffectiveness, leading to an increase in vulnerability in fighting infections, as well as the prevalence of autoimmune diseases [15,16,17]. However, this concern is divided within the community, as some believe that autoimmune disease prevalence does not increase with age. Using this as a starting point, we can turn to the differences in immunity across age cohorts to better understand the aging immune system.

The prevalence of neonatal autoimmune diseases is rare, although newborns have a relatively undeveloped immune system. Compared to their developed adult counterparts, newborns secrete fewer cytokines, have impaired antigen secretion, and have poor functional T lymphocytes [18]. Neutrophils, which are the first line of defense in the weakened innate immune system of newborns, display a poor response to pro-inflammatory cascades [16]. T-cells also differ greatly between neonates and adults, with the quantity of CD4^+^ cells and naïve T-cells being high during birth and continuously decreasing to their final amount by the age of 5 [16]. The amount of pro-inflammatory cytokines such as IL-ß and TNF-a increases rapidly and begins to settle to a balanced quantity around the age of 1 [19]. However, the efficiency of newborn immune systems is overall impaired compared to that of young adults and adults due to immature T-cell antigens.

As age progresses, the immune system becomes stronger and is able to prevent and counteract a wide variety of illnesses. While innate immunity is what initially begins to decline as we age, there are significant alterations to adaptive immunity as well. Over the age of 80, the number of combined organ-specific and non-specific auto-antibodies is 7-times that of normal adults, which is around 2% [20]. The memory of T and B cells expands from being exposed to both vaccines and previous antigens over the years. However, at older ages, some argue a decrease in the efficacy of T and B cells makes the aging population especially susceptible to infections and inflammation. During the senescence process, as previously mentioned, there is an increase in memory T cells and a decrease in naïve T-cells into adulthood. This reduces the body’s efficiency in creating an abundance of new antigens, further reducing the antigen response [15]. There is a decrease in the efficiency of recycling and removing damaged cells from disrupted metabolic pathways and pro-inflammatory cytokines including IL-1ß, IL-18, and TNF-a, which could account for an increase in autoimmune disease prevalence [16,20,21].

Given these characteristics of the aging immune system, it will be critical to identify the factors modulating the effect of MSCs on memory T cells to improve therapeutic efficacy in aging recipients. In addition, characterizing the response of MSCs to inflammaging cytokines such as IL-1ß, IL-18, and TNF-a will allow us to understand and ultimately predict the function of MSCs in older patients. Key parameters relating to memory T cell quantity and activity, metabolic alterations and pro-inflammatory cytokines can be developed as biomarkers to aid in the matching of donor MSCs to the immune environment of the recipient.

### 2.2. Aging of MSCs

Senescence extends beyond the immune systems, as there have been observations indicating the diminished effects of MSC in the elderly. In short, MSCs of the elderly exhibit abnormalities in cell differentiation, have diminished viability and proliferative potential, and have morphological changes indicative of functional defects. When comparing 30–40-year-old bone marrow-derived MSCs with those of patients aged 60–80 years old, there was a decrease in proliferative and osteogenic potentials [22,23,24]. In addition, in vitro cell culture through extended passages was found to stimulate controlled aging processes and change MSC morphology. For example, the cells lost the characteristic spindle shape of fibroblasts. When further expanded, MSCs had compromised osteogenic potential, while they maintained their ability to differentiate towards adipose lineages [25].

During the aging process, one senescent cell can accelerate the senescence and aging process of surrounding cells, and MSCs can influence the microenvironment of cells by releasing reactive oxidative species such as nitrous oxides [24,26,27]. It has been shown that MSCs from older donors have a reduced ability to catalyze superoxide radicals while having an increased susceptibility to producing nitrous oxides and reactive oxidative species [23,28]. Young cells, harvested from patients under 40 years old, compared to cells extracted from people over 50, were found to have a two-fold antioxidant defense mechanism [28,29,30]. Decreased defense against oxidative stress contributes to increased apoptosis and possibly aged cells, producing smaller colonies of cells, as shown by a decrease in cells/gm with increasing age [28]. While reactive oxygen species have been shown in some studies to increase, overall, this process is not well understood concerning how stress accumulates in MSCs and the downregulated biological cascades that occur as a result.

From MSC transplantation studies, it has also been found that the age of the donor plays an important role. Rats with central nervous system demyelination had bone marrow-derived MSC (BM-MSC) treatment from either young or old donors, and in this model young BM-MSCs were able to migrate to the lesions and differentiate into neuronal cells in recipients more efficiently than old BM-MSCs [31,32].

Based on these lines of evidence, cryopreservation of young MSCs has been proposed to maximize the efficacy of MSC therapy. However, this approach has not been tested on a large scale. It should also be mentioned that some studies have found that at least certain MSC properties are maintained in aged donors [33,34]. Therefore, further studies are required to pinpoint exact differences between aged and young MSCs.

## 3. Impact of Sex as a Biological Variable on the Immune System and MSCs

The immune system is shaped by a variety of endogenous and exogenous factors, which sexual dimorphisms contribute to. Differences in sex are central to immune function and the responses the body has to stimuli. In addition to the fact that female dendritic cells produce more IFN-a, women have greater activity and counts of neutrophils and macrophages than their counterparts. In males, a surge in pro-inflammatory cytokines (IL-6 and TNF-a) and increased activity of NK cells have been found [35,36]. Females demonstrate stronger cellular innate and humoral immune responses, supporting a greater resistance to infections than males [37]. Due to an intensified innate immune response, females, unlike their counterparts, are more prone to autoimmune diseases [38]. Estrogen, produced in high quantities in women during the ovarian cycle, suppresses the innate immune system by the diminished ability to produce IL-1B, IL-6, and TNF by monocytes. In addition to the roles of sex hormones, the central differences in the immune system are also grounded in the gene composition of the X chromosome [39]. X-linked gene dosage has been demonstrated to affect autoimmune disease risk, and recently skewed expression in X-linked genes has been found to associate with female-biased autoimmune diseases, including Sjögren’s syndrome and systemic lupus erythematosus [40,41]. Regardless of the mounting evidence showing the important role of sex, there are, overall, a limited number of clinical trials and experimental models that consider biological sex as a variable [37].

In addition to sex differences in the recipient immune system, sex differences in donor MSCs have been described in both animal and human cells. The higher expression of Rankl and Opg genes in female murine osteoblasts increased the number of osteoblasts created post MSC differentiation [42,43]. Adipose-derived MSCs derived from male piglets produced IL-6 at greater quantities compared to female cells [44]. A biostatistical analysis comparing chromosomal segments of genes from male and female adipose-derived MSC donor cells proposed that stem cell differentiation and migration is differentially regulated between the two sexes. Specifically, 35 out of 40 of significantly regulated genes in MSCs were attributed to sex-based differences, with 20 genes overexpressed in males. Functionally, the lower expression of CXCL3 in male MSCs may underlie their reduced capability to differentiate into adipocytes [45].

While the number of experiments specifically investigating sexual dimorphisms is on the rise, our understanding on the effect of sex as a biological variable on MSCs and their interaction with the immune system is not yet fully understood. Future studies are warranted to establish whether donor/recipient sex provides variability to the regulatory and protective functions of MSCs, and if future therapies will allow cross-sex transplantation.

## 4. Impact of Biological Source on MSC Autoimmune Therapy

As discussed previously, one major application of MSCs is their therapeutic use in autoimmune diseases. With many clinical trials underway and more about to start, it raises the question of whether it is possible to use MSCs from one biological source for various autoimmune diseases, depending on the overlap of molecular and cellular alterations in the diseases. In addition, MSCs can be readily derived from multiple sources, including the bone marrow, adipose tissue and umbilical cord, and MSCs from different sources may have varying biological properties. Therefore, it is important to understand the impact of the biological source on the outcome of MSC therapy for different autoimmune diseases in order to guide clinical decisions.

### 4.1. Osteoarthritis (OA)

The use of MSCs has become a driving force in better combating autoimmunity-caused cartilage breakdown, such as that in osteoarthritis (OA). OA is one of the most common joint diseases as it affects 20% of the adult and aging population and is predicted to affect 67 million people by 2030 [46]. OA is further subdivided into two groups: primary and secondary. Primary OA is hereditary and gene-dependent, while secondary OA is due to pre-existing conditions and trauma to the joints and cartilage [47]. The characteristics of OA include loss of articular cartilage, thickening of the subchondral bone, and increased joint stiffness, in addition to swelling and pain [46,48]. The molecular pathology of osteoarthritis remains largely unknown. However, a multitude of factors contribute to cartilage deterioration, such as genetic factors, age, and obesity. Recent research suggests that transforming growth factor (TGF) and signaling molecules are involved in the transition from homeostatic joint function to a catabolic state [49]. TGF-downregulation causes matrix metalloproteinase expression to degrade the structural proteins that make up cartilage.

Until recently, OA has not been considered to be an inflammatory disease, but there is indication that inflammation of the synovial tissue and the immune system is involved. More specifically, TNF-a, IL-1, and IL-6 have been shown in increased concentrations in patients with OA, yet the connection to biological disease progression and various secretions of macrophages and T cells is not well understood [50,51,52]. OA has also been characterized as a mesenchymal disease; as in, MSCs have a reduced rate of proliferation and often fail at differentiation and mobilization, making them unable to regenerate the necessary degraded cartilage [53].

Current progression of treatment has been limited and rooted in reducing pain, rather than targeting and understanding the molecular and cellular phenomena of OA pathology. Treatment options include both steroid and non-steroid anti-inflammatory drugs, injections of sodium hyaluronate, and omega-3 fatty acids [54]. A possible solution moving forward is to use MSCs derived from multiple locations throughout the body, including the synovial membrane of OA patients, to regenerate joint and cartilage function.

If animal trials are any indication of the therapeutic effects of MSCs, the future of therapeutic advances could be promising. In sheep models, a reduction in biomarkers PGE2, TNF-a, and TGF-B, with an increase in aggrecan, the main component of cartilage, was observed post MSC transplantation [55]. To date, there have been a handful of clinical trials assessing the therapeutic potential of MSCs in humans, which we are reviewing below.

#### 4.1.1. Bone Marrow-Derived MSCs (BM-MSCs) in OA

One notable clinical trial using bone marrow derived MSC’s was conducted by Shapiro et al. in 2016 with a cohort of 25 patients experiencing bilateral OA pain. Specifically, bone marrow derived concentrated serum, which has a large concentration of MSC’s, was injected into one of the patient’s knees while the other knee had a saline placebo injection. Patients found that there was a diminished effect of pain in the knee injected with the bone marrow concentrate, possibly explained by MSC’s involvement in decreasing inflammation and their role as signaling cells [46,56].

Bone marrow-derived MSCs are particularly useful in OA due to their ability to maintain chondrogenic differentiation and the secretion of biomolecules that form the extracellular matrix in joints. In comparison to direct chondrocyte transplantation, MSC transplantation was found to be just as effective, while using less invasive techniques, and overall being a more cost-effective procedure [57]. Chahal et al. also performed a bone marrow derived transplant with 12 patients that received intra-articular injections of various concentrations of MSCs. Patients receiving greater MSC numbers had significantly lower cartilage catabolic markers and inflammatory cytokines (IL12p40) and an increased level of angiogenic factor, vascular endothelial growth factor (VEGF) than those receiving little to no MSCs [58].

#### 4.1.2. Adipose-Derived MSCs (ASCs) in OA

Another form of effective therapy that has been used on patients suffering from OA is proliferating MSCs derived from adipose tissue. Adipose-derived MSCs (ASCs) have similar biological properties to bone marrow-derived cells; however, unlike bone marrow-derived MSCs, they are less invasive and painful for the patient to obtain. In addition to the increased accessibility, ASCs have shorter doubling time in culture, can be cultured for a greater number of passages, are more resistant to senescence, have longer retention of stem cell characteristics, and are easier to cryopreserve than bone marrow-derived stem cells [59]. Their ease of accessibility and overall, less painful procedure makes the use of ASCs a viable and appealing option for autoimmune therapies. Indeed, ASCs have been found to be a safe alternative to reliving knee pain in OA. When using the Western Ontario and McMaster University Osteoarthritis Index (WOMAC) to quantify OA pain, patients that had low-dose ASC injections had a score reduction of 30.7 10.7 mm compared to a decrease of 22.0 1.4 mm when using the current hyaluronic acid treatment [60,61].

In a separate study, it has been shown that increasing the number of ASCs injected into the intra-articular space had better outcomes associated with decreasing knee pain. Patients within the high-dose experimental group had a reduction in pain in 39% compared to the initial baseline and a WOMAC score drop from 54.2 ± 5.2 to 32.8 ± 6.3, over the course of 6 months, while patients that were given low-to-medium doses of ASCs reported little to no improvement overtime [62]. These studies encourage future trials to determine the efficacy of adipose-derived stem cells for OA, as most human and animal trials have so far primarily centered around bone marrow-derived MSCs.

#### 4.1.3. Umbilical-Cord Derived MSCs (UMC-MSCs) in OA

The final form of MSC therapy that has been implemented in both animal and human clinical trials utilizes umbilical cord MSCs (UMC-MSCs). Since January 2021, there have been six clinical trials around the world utilizing these stem cells for OA. Unlike BM and ASCs, UMCs provide fewer cell colonies in addition to having smaller yields when compared [52]. Nonetheless, from the human clinical trials that implemented them, UMC-MSCs also provided improvement for OA patients. During a yearlong trial with extensive follow-ups, patients in the experimental group who received a knee intra-articular injection of 2–3 × 10^7^ cells reported no reoccurring knee pain. Additionally, WOMAC scores throughout treatment decreased. From this clinical trial, the main findings that point to further use of UMC-MSCs are that reductions in symptoms take effect 1 month after treatment and can last, as in this case, for 6 months [63].

Others performed multiple doses of injection, each comprising 20 × 10^6^ cells into the intra-articular space and noted that WOMAC scores of patients that had repeated dosages were significantly lower and the pain did not return as quickly. When the joints were imaged using magnetic resonance technology, there was no difference between any of the treatment groups, using MSCs or the traditional treatment methods [64]. Further research needs to be conducted on the role of various types of MSCs and their uses in OA. A possible route for future studies utilizes multiple rounds of MSC treatments, exposing patients to a continuous supply of MSCs for a set treatment period.

### 4.2. Multiple Sclerosis

Multiple sclerosis (MS) is at the forefront of neuroimmunology, closing the gap between inflammatory autoimmune diseases and demyelinating neurological conditions. The pathogenesis of MS centers around demyelination of the myelin sheath in the brain and spinal cord, resulting in lesions of both white and gray matter [23]. The lesions manifest in both physical and clinical decline, as presented by tremors, vision problems, and problems remembering information. Around the world, around 2,000,000 people suffer from MS, of which about half a million suffer in the United States alone. While men are more commonly diagnosed with MS, the likelihood of women having MS is much greater [24,25]. The direct cause of MS remains largely unknown, as signs point to a contribution from factors ranging from genetic, environmental, and a cascade of immunological pathways. Interleukin-2 receptor alpha genes (IL2RA) and interleukin-7 receptor alpha genes (IL7RA) are thought to contribute to an increased risk of MS in people [65,66]. As an autoimmune disease, myelin antigens are a target for autoreactive T lymphocytes and disease progression is triggered by T helper 17 (Th17) and T helper 1 (Th1) cells. Common proinflammatory cytokines such as IL-1B and TNF-a have also been commonly thought to contribute to the advancement of MS [31,67].

Current clinical treatments of administering interferons, glatiramer acetate, and mitoxantrone focus on subsiding disease manifestations rather than being able to reverse white and gray matter lesions. Dangerous and intolerable treatment options raise the possibility of using autologous and allogenic MSCs as potential therapeutic options. This has shown promising results in both animal and clinical trials around the world [68]. According to Clinicaltrials.org, there have been 29 medical trials that have been actively conducted around the world focusing on MSCs in hopes of understanding and regulating MS. MSC transplantation and stem cell therapies have been shown to improve central nervous system function and aid in the regeneration of brain lesions. In addition to a few MSCs exhibiting astrocyte markers, MSCs in the brain have also been observed to differentiate into oligodendrocytes [31]. These characteristics help support the notion of possible therapies. However, with MS, the number of preclinical studies using animal models is far greater than the clinical trials occurring around the world, with common murine models involving induced encephalomyelitis, a manifestation of early onset clinical MS [31]. In this section, we will be summarizing the effects of BM-MSCs, ASCs, and UMC-MSCs on MS disease progression, primarily in animal studies, which may provide insights into its future translation into humans.

#### 4.2.1. BM-MSCs in MS

As BM-MSCs were the first to be discovered, the majority of the studies conducted by far have utilized these cells. BM-MSCs have been shown to improve neurological symptoms and reduce pro-inflammatory cytokines, along with decreasing demyelination associated with MS, due to their ability to decrease stress inflicted on the blood–brain barrier. This improvement in blood–brain barrier function is thought to be, in part, due to a decrease in the levels of the pro-inflammatory cytokines IL-6 and TNF-a, and an increase in the levels of the anti-inflammatory cytokine IL-10 [31,69,70]. The effectiveness of BM-MSCs is thought to result, in part, from their capabilities to prevent apoptosis post differentiation [31].

Similarly, when human BM-MSCs were injected intravenously into mice with encephalomyelitis, there were improvements in clinical scores and significant decreases in levels of pro-inflammatory cytokines such as IL-2, IFN-γ, IL-12p70, and TNF-a [71]. Intriguingly, it has been shown that BM-MSCs within the nervous system can localize to damaged areas and repair the damage via varying hypothesized mechanisms, one of which points to their production of neurotrophic and neuroprotective factors. In addition, BM-MSCs suppress T lymphocytes by direct cell-to-cell contact and subsequently alter MS progression. Localized movement of BM-MSCs associate with improved hypothalamus function in mice with hypothalamus damage, concurrent with an increase in differentiation into glial cells and oligodendrocytes. More specifically, there was a 14% increase in recorded BM-MSC cultures when compared to control groups, and 2 of the 8 mice that received treatment had a gait from recovering from hind limb paralysis [71]. The ability of BM-MSCs to migrate to damaged parts of the brain and differentiate into neuronal and surrounding cells could be a distinguishing factor of MSCs and offer future insights into how repair or immune suppression occurs.

#### 4.2.2. ASCs in MS

Given that ASCs are much more accessible and require a less invasive procedure, it is worthwhile to investigate the differences in biological markers that may distinguish them from BM-MSCs. When comparing the two most commonly used MSCs, cytokine secretion of IL-8, IL-6, and TNF-a, together with their differentiation abilities, are similar [72]. Using human ASCs, similar to BM-MSCs, there was an overall decrease in pro-inflammatory cytokine IL-17 and an increase in the production of anti-inflammatory cytokines IL-4 and IL-10 in MS models [31,73,74]. In a clinical trial involving 34 patients, ASCs proved to be safe as patients experienced minor symptoms involving urinary tract infections, yet provided lackluster neurological protection in reoccurring MS [75]. As the number of studies utilizing ASCs are currently smaller than those utilizing BM-MSCs, future research in this area will shed light on the clinical choice of MSCs in MS [76].

#### 4.2.3. UMC-MSCs in MS

The increased use of UMC-MSCs across disciplines provides therapeutic promise in MS. Unlike BM-MSCs, UMC-MSCs are not limited by the donor’s age and physical condition, while retaining the ability to differentiate, regulate the immune system, and repair the nervous system [77,78]. Few studies have noted the initial impact and biochemical changes that occur after UMC-MSC transplantation. To analyze these changes, a non-human primate model using the cynomolgus monkey, which has a similar immune system to humans and has exemplified strong symptoms of encephalomyelitis, was used [79]. Three monkeys out of six that had encephalomyelitis were treated with UMC-MSCs and exhibited a decrease in demyelinated regions after two treatments when observed by transmission electron microscopy. After treatment, the levels of IL-5, IL-17A and IFNs significantly declined and those of IL-4 and IL-10 significantly increased from the initial stages of disease progression. This study did not find a decrease in demyelinated white matter in MRI images; however, there were improvements in clinical manifestations [78].

In a clinical trial where patients were exposed to a combination of either anti-inflammatory/immunosuppressants and UMC-MSCs, or just administered anti-inflammatory or immunosuppressive medicine. Patients in the UMC-MSC group had fewer relapses and disease progression that was more consistent and smoother during the 1-year observation period. Levels of IL-4 and IL-10 in the MSC treatment group were significantly increased through the treatment period, while TNF-a and IL-17 decreased. In contrast, the levels of these cytokines did not change significantly in the control group [80]. Similar results were observed in a separate clinical trial, in which overall pro-inflammatory cytokines decreased while anti-inflammatory cytokines increased with MSC therapy [81,82]. The beneficial results of MSC treatment demonstrated in various species, and the less invasive nature of their associated procedures provide great promise for UMC-MSC therapies.

### 4.3. Systemic Lupus Erythematosus (SLE)

The Lupus Foundation estimates that there are around 1.5 million people in the United States and 5 million in the world that suffer from a form of lupus. It is estimated that 9 out of 10 people diagnosed with lupus are women. The direct causes of lupus are not yet completely understood, but it has been hypothesized that genetics, the environment, hormone imbalances, and immunoregulatory process all impact someone’s chances of being diagnosed with lupus [83]. Systemic lupus erythematous (SLE) is one of the most common forms of lupus and it causes deterioration and inflammation of the kidneys, joints, skin, and muscles [84,85]. SLE is an autoimmune disease that is molecularly activated by the increased production of monocytes, lymphocytes and autoantibody production caused by failure of the innate and adaptive immune system [86,87]. Defects in regulatory T cells (Treg) are thought to disrupt immune homeostasis and promote autoimmune processes in SLE, with the decrease in Treg levels in patients with SLE leading to a decrease in IL-10 [86,88]. Additionally, individuals with SLE have been reported to have higher levels of IL-6, IL-12, IL-17, IFN-γ and lower levels of IL-4 compared with healthy controls [89].

Current SLE treatments include non-steroid anti-inflammatory drugs and corticosteroids, which provide short-term relief but have severe adverse reactions of increased toxicity and generalized immune suppression [85]. For example, corticosteroids in combination with cyclophosphamide, which is a standard of care, leads to steroid-induced diabetes, cataracts, and gonadal toxicity [90,91,92].

Regarding MSC therapy in SLE, notably, it has been found that MSCs from SLE patients are deficient in immunosuppressive and anti-inflammatory functions. Primary observations of MSCs from SLE patients showed an increase in senescence and slowed proliferation compared to healthy counterparts. SLE BM-MSCs could only be cultured for around 5–10 passages before slowing down and exhibiting deficiencies, while normal BM-MSC could be cultured for up 40 passages consistently without showing a significant decrease in vitality [93]. In addition, the cytoskeleton of BM-MSCs was abnormal and associated with impaired function of SLE-MSCs [88]. While both healthy and diseased MSCs exhibited the ability to release IL-7, IL-6, and IL-11, defects in SLE MSCs argue against the use of autologous transfer for disease treatment, as further discussed below.

#### 4.3.1. BM-MSCs in SLE

In a small clinical trial, two female SLE patients below the age of 26 received autologous MSC treatment using BM-MSCs. After various follow-up consultations, it was found that there were no adverse effects resulting from the transplantation. However, there was no significant improvement in disease activity measured by SLEDAI scores. This was surprising because there was an increased CD4^+^CD25^+^FoxP3^+^ T regulatory (Treg) cell counts in patients, and in vitro, the MSCs suppressed the proliferation and activation of normal peripheral blood lymphocytes [94]. In a separate study that involved the transplantation of BM-MSCs derived from healthy donors, most of the 35 enrolled SLE patients experienced significant improvements in blood cell count in parallel with the decline in disease activity, increased Treg and decreased Th17 counts. No adverse events related to transplantation were observed [95].

#### 4.3.2. ASCs in SLE

The majority of experimental murine models and clinical trials for SLE therapies involve the use of either BM or UMC-derived MSC. Adipose-derived MSC models are far less common, yet their use could be overlooked. Transplantation of human ASCs into Roquin (san/san) mice presenting a lupus phenotype increased the levels of B regulatory cells (Bregs) and Foxp3-expressing regulatory T cells [96]. This experiment illustrates the potential effects of using ASCs to target both B and T cell pathologies in SLE.

#### 4.3.3. UMC-MSCs in SLE

As mentioned in the previous section, bone marrow-derived MSCs currently comprise the majority of leading stem cell therapy trials and experiments, while their collection is invasive. Keeping this in mind, umbilical-cord-derived stem cells might provide the most therapeutic relief and most viable option for SLE treatment. In a study using the MRL/lpr model, mice were broken down into various experimental cohorts receiving UMC-MSC transplantation, BM-MSC transplantation, or vehicle treatment, respectively. Mice receiving three transfusions compared to those without any or one transfusion saw better outcomes and regression of SLE symptoms, exemplified by decreased urine protein levels, reduced monocyte chemotactic protein-1 (MCP-1) levels, and improved renal formation. Mechanistically, there was an upregulation of Treg cells in mice receiving MSCs. Notably, this experiment showed that the outcome of UMC-MSC transplantation is similar to that of BM-MSC, and occurrence and frequency of treatment plays a considerable role [88]. Additionally, in a clinical trial of 30 individuals with refractory SLE, it was noted that cell transplantation was safe with minimal side effects and there was an increase in Treg cell account, accompanied by a consistent decrease in IL-17 levels at 3, 6, and 12 months post-treatment. In addition, IL-6 levels increased throughout the follow-ups and there was also a rapid increase in TGF-B and a settling out by the 12-month mark. The cytokine concentration was not dependent of the doses of cells administered to the patients [97]. These results are consistent with those observed in BM-MSC studies.

### 4.4. Comparison of MSCs from Different Sources across Autoimmune Diseases

In previous sections, we reviewed the use of MSCs from different biological sources, including bone marrow, adipose, and umbilical cord, in various autoimmune diseases. Here, by summarizing the main findings in a table (Table 1), we note that both BM-MSCs and UMC-MSCs show consistent effectiveness across diseases. In addition to improvement in disease scores, the administration of BM-MSCs and UMC-MSCs commonly lead to an increase in levels of anti-inflammatory cells or cytokines and a decrease in levels of pro-inflammatory cells or cytokines (Table 1). The exact panel of cell types and cytokines being evaluated in each study varies, however, precluding direct comparison. The future establishment of a standardized assessment panel may aid in the development of biomarkers that can guide the choice of MSCs based on the immune milieu of the recipient; correspondingly, if a common biomarker can be found across patients, that would argue for the choice of a “universal MSC donor”.

## 5. Conclusions

Mesenchymal stem/stromal cells (MSCs) hold great promise for the treatment of autoimmune conditions given their immunomodulatory properties. The low immunogenicity of MSCs has spurred the investigation into the potential use of MSCs from a “universal donor”, which allows the expansion of MSCs, both in vivo and in vitro, for immediate and standardized care for autoimmune disease patients.

With the perspective of “universal” MSCs, to establish the criteria for what constitutes a suitable donor, we should turn to some of the findings referenced above. Ideally, autologous MSCs are the choice of safety as allogenic transplantation could lead to cell rejection. However, cells obtained from patients suffering from autoimmune diseases may behave differently than those from healthy donors, including deficiency in the ability to proliferate and successfully differentiate. To this end, it is preferable that MSCs be obtained from BMs from young healthy donors or derived from UMCs directly after birth from both cesarian and virginal births. While more research is needed to elucidate the role of sex in MSC therapeutic ability, current studies suggest fundamental differences in immune regulation between men and women. Therefore, sex as a biological variable should not be neglected in MSC transplantation. Based on these findings, one should use MSCs that (1) are immunologically tolerated in the recipient, (2) do not show signs of cellular senescence, including compromised proliferation and differentiation capabilities, (3) compatible with the biological sex of the recipient in regard to sex-specific immune processes, and (4) are known to be effective in attenuating hyperactivated cytokine and immune cell activities in the recipient from clinical studies for transplantation. Using these basic criteria, we have proposed that it is worthwhile to further establish MSC banks from multiple donors that span a range of biological ages, tissue sources and sex for the selection of future transplantation therapies.

In this review, we summarized findings, highlighting the impact of age, sex, and biological sources on the function of MSCs. The fact that MSC function is significantly impacted by these biological variables indicates that certain levels of personalization of MSC therapy may be desired. However, common themes have arisen from these studies, such as the preferable use of young MSCs over old, as well as the effectiveness of MSCs in reducing IL-17 levels and increasing IL-10 levels. While this review focused on the use of MSCs in autoimmune diseases, we recognize that MSCs’ immunomodulatory properties allow for their broad use in other treatments. One such use that has been rapidly expanding has been the use of MSCs on patients recovering from COVID-19. Similar to the use of MSCs in autoimmune diseases, those suffering from COVID-19 had no adverse effects from cell transplantation and also had improved lung function, likely resulting from an attenuation of the cytokine storm post SARS-CoV-2 infection [98,99]. Therefore, it may be feasible to ultimately develop a standardized MSC therapy for subgroups of patients, as more studies involving cohorts with defined age and sex information, as well as standardized molecular and cellular measurements, emerge.

## Figures and Tables

**Table 1 cells-11-02077-t001:** Summary of the effectiveness and biomarkers found by utilizing bone marrow (BM-MSC), adipose (AD-MSC), and umbilical-cord (UMC-MSC)-derived MSCs in osteoarthritis, multiple sclerosis, and systemic lupus erythematous.

		Osteoarthritis	Multiple Sclerosis	Lupus
**Bone-marrow-derived MSC**	Effectiveness	Knee pain reduction	Reduce microgliosis and astrocytosis Increase BBB function Increased oligodendrocytes	Suppressed in vitro peripheral blood lymphocyte levels Improved blood cell count
	Biomarker	IL-12p40 decreases VEGF increases	IL-10 increased IL-4 increased IL-6 increase Glutathione increased IL-6 decreased IL-1ß decreased TNF-α decreased IL-12p70 decreased VEGF increase	Increased CD4^+^CD25^+^FoxP3^+^ cell counts Treg increased Th17 decreased
**Adipose-derived MSC**	Effectiveness	Strengthens joints Decreased WOMAC scores Increased synovial lining	Less effective Increased symptoms of urinary tract infections Temporarily increased severity of MS then decreased	Reduction of SLEDAI scores Lower than baseline of urine proteins Increased renal function
	Biomarker	VEGF increase TGF-ß secretion	IL-10 increased IL-4 increased IL-17 decrease Inhibited T-cell expansion	Breg ncreased Foxp3-expressing regulatory T cells increased
**Umbilical-cord-derived MSC**	Effectiveness	No reoccurring knee pain Decreased WOMAC scores	Demyelinated region did not decrease on MRI Promoted remyelination Clinical manifestations improved and less relapses Reduced astrogliosis	Improved renal function Reduction of SLEDAI scores
	Biomarker		IL-10 increased IL-4 increased IL-5 decreased TNF-α decreased IL-17 decrease HGF increase VEGF increases Decreased NK cells	MCP-1 decreased in mice Urine proteins decrease Treg increased Inhibited Th17 cells IL-17 decreased No changes in IL-6 nor IL-17A TNF-a decreased

## Data Availability

Not applicable.

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
