# Peer review of "Universal or Personalized Mesenchymal Stem Cell Therapies: Impact of Age, Sex, and Biological Source"

_cells, 2022, doi:10.3390/cells11132077_

Round 1

Reviewer 1 Report

Carp and Liang in their review “Universal or Personalized Mesenchymal Stem Cell Therapies: Impact of Age, Sex, and Biological Source” in a very detailed and comprehensive way report on the use of MSCs from different tissue sources (BM, AD-MSCs, UMC-derived MSCs) in a series of autoimmune diseases such as OA, MS, SLE.

This is a very comprehensive and well-written review on the use of MSCs in autoimmune diseases and factors that influence this therapeutic effect. However, what I miss in this review are

1.     Exactly defined criteria for selection of personal or “universal” donors for MSCs

2.     According to the authors, what is the solution for this issue, say establishing of the MSC-banks from multiple donors of bone marrow, or any other tissue source for MSCs.

Author Response

We thank the reviewer for the comments. Please see our response below -

1. Exactly defined criteria for selection of personal or “universal” donors for MSCs

Response: We have now clearly defined the criteria for the selection of MSC donors in 'Conclusion'. Donor MSCs that meet the criteria include those that 1) are immunologically tolerated in recipient, 2) do not show signs of cellular senescence including compromised proliferation and differentiation capabilities, 3) compatible with the biological sex of the recipient in regard to sex-specific immune processes, and 4) are known to be effective in attenuating hyperactivated cytokine and immune cell activities in recipient from clinical studies.

2. According to the authors, what is the solution for this issue, say establishing of the MSC-banks from multiple donors of bone marrow, or any other tissue source for MSCs.

Response: We have now proposed a solution for this issue in 'Conclusion'. We propose that it is worthwile to further establish MSC banks from multiple donors that span a range of biological age, tissue source and sex for the selection of future transplantation therapies

Reviewer 2 Report

The authors took an interesting approach to the promise of using MSCs for "Universal" use and discuss possible interferences in the quality of these cells, such as aging and donor sex, the origin of the cells, and also different types of autoimmune diseases. However, I resent the lack of a more objective and conclusive analysis of data. It was missed current reports for the autoimmune storms in Covid-19.

Author Response

We thank the reviewer for the comments. We have now added a discussion on reports for the autoimmune storms in COVID-19 in 'Conclusion' - 'While this review focused on the use of MSCs in autoimmune diseases, we recognize that MSCs’ immunomodulatory properties allow for their broad use in other treatments. One such use that has been rapidly expanding has been the use of MSCs on patients recovering from COVID-19. Similar to the use of MSCs in autoimmune diseases, those suffering from COVID-19 had no adverse effects from cell transplantation and also had improved lung function, likely resulting from an attenuation of the cytokine storm post SARS-CoV-2 infection.'

Round 2

Reviewer 2 Report

The proposed review fulfilled its objective. However, I observe that with the systematic review, this could occur with more assertiveness and potential for use in guidelines.